# Conceptual Framework of the Design of Pleiotropic Drugs against Alzheimer’s Disease

**DOI:** 10.3390/pharmaceutics15102382

**Published:** 2023-09-26

**Authors:** Thomas Guiselin, Cédric Lecoutey, Christophe Rochais, Patrick Dallemagne

**Affiliations:** Normandie University, Unicaen, Centre d’Etudes et de Recherche sur le Médicament de Normandie (CERMN), 14000 Caen, France; thomas.guiselin@unicaen.fr (T.G.); cedric.lecoutey@unicaen.fr (C.L.); christophe.rochais@unicaen.fr (C.R.)

**Keywords:** drug, pleiotropic, neurodegeneration, Alzheimer’s disease

## Abstract

The multifactorial nature of some diseases, particularly neurodegenerative diseases such as Alzheimer’s disease, frequently requires the use of several drugs. These drug cocktails are not without drawbacks in terms of increased adverse effects, drug–drug interactions or low adherence to treatment. The use of pleiotropic drugs, which combine, within a single molecule, several activities directed against distinct therapeutic targets, makes it possible to overcome some of these problems. In addition, these pleiotropic drugs generally lead to the expression of a synergy of effects, sometimes greater than that observed with a combination of drugs. This article will review, through recent examples, the different kinds of pleiotropic drugs being studied or already present on the market of medicines, with a focus on the structural aspect of such drug design.

## 1. The Concept of Pleiotropic Active Drugs

Since Paracelsus and the laborious beginning of the extraction of active ingredients from materia medica, which finally occurred only three centuries later with the extraction of morphine, people, in order to explain the therapeutic efficacy of medicines, have gradually imagined the notion of biological targets. 

Theorised by Paul Ehrlich and his famous “magic bullet” at the beginning of the 20th century [1], this concept has been progressively enriched by the notion of selectivity of the active principle, with regard to a single target, in order to avoid possible deleterious side effects as much as possible. Although this selectivity has been sought for a long time, it has only recently and rarely been achieved, and most of today’s active therapeutic compounds still aim at a main target that is responsible for the desired effect; unfortunately, they may also modulate secondary targets (off-targets) that are responsible for side effects (Figure 1A). The notable exception to this lack of selectivity is the therapeutic use of antibodies, whether naked or armed with an active principle, a radioelement or even a toxin, which show absolute selectivity towards their target, a specific antigen towards which they are directed. However, this notion of selectivity is evolving [2] and, in particular, it is being dissociated from targeted therapy, which should nowadays be described as personalised therapy. The effectiveness of selective therapies sometimes comes up against certain diseases of multifactorial origin for which the use of drug combinations is recommended. This is particularly the case in oncology, where multi-drug therapy is the rule, but it is also the case in virology, where the famous triple therapy has considerably improved the management of AIDS patients. The benefit sought in this case is both to obtain greater efficacy than that of a single drug, thanks to a synergistic effect, and also to combat resistance phenomena. However, the combination of several pharmaceutical forms is not without disadvantages, particularly in terms of increased side effects, drug–drug interactions and also low compliance with treatment. These disadvantages have led to the development of so-called “combo” drugs, which combine several active ingredients in a single pharmaceutical form. This is the case for many antihypertensive, antiretroviral or anti-Alzheimer’s drugs. However, these combos do not solve the problems of toxicity and interactions.

In order to answer these questions, the concept of so-called pleiotropic drugs has been submitted [3]. A pleiotropic compound (Multi-Target Directed Ligand) has been specifically designed to express effects on several targets, which are all of therapeutic interest. This multiple action should allow it to exert a synergy of desired effects “squared” that leads, in theory, to a clinical efficacy superior to that obtained with an ordinary drug and even with a combination of ordinary drugs (Figure 1B). Such a pleiotropic compound is the result of a rigorous design and should not be confused with a ubiquitous or promiscuous drug, whose lack of selectivity has not been reasoned out and which generally only leads to undesirable effects in the clinic. In short, according to the concept of pleiotropic compounds, the undesirable side effects of ordinary active ingredients become therapeutic, and indeed these agents with multiple effects are often more devoid of deleterious effects than ordinary active ingredients. There are two reasons for this: on the one hand, the fact that a synergy is obtained makes it possible to reduce the active concentration administered and thus to minimise the adverse effects and, on the other hand, there is a saturability of the ligands. Each biologically active molecule can statistically only show a high affinity towards a limited number of therapeutic proteins. If all these targets are of therapeutic interest, this could reduce the risk of unwanted effects caused by interactions with undesirable targets. In addition to this increased efficacy and reduction in adverse effects, pleiotropic substances can also be credited with avoiding the problems of interactions and low compliance encountered with multiple-medication therapy.

Pleiotropic compounds have already begun to take over our pharmacopoeia, particularly in the class of anticancer drugs, among which, many kinase inhibitors have multiple effects by targeting several enzymes, some of which are even described as pan-inhibitors, such as pazopanib, which is prescribed for metastatic breast cancer. However, its pleiotropic nature does not prevent it from remaining a targeted or rather personalised therapy drug, given the possibility of selecting patients whose tumour status corresponds to the kinases targeted by the drug. Neurodegenerative diseases are another area where a pleiotropic therapeutic intervention seems necessary.

## 2. The Application of the Concept of Pleiotropic Active Compounds to the Therapeutic Management of Alzheimer’s Disease

Given the multifactorial nature of their pathogenicity, neurodegenerative diseases, such as Alzheimer’s disease, appear to be able to respond positively to treatment with pleiotropic compounds, provided that they can be administered precociously through the early diagnostic stage which still needs to be improved. Some studies have recently focused on this approach [4,5,6]. Overall, the literature reports four types of pleiotropic agents designed for the potential management of Alzheimer’s disease (Figure 2).

### 2.1. Conjugates with Immolable Chemical Bond


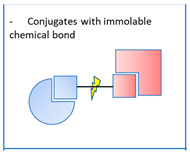
. Also known as mutual prodrugs or codrugs [7,8,9], these molecules have two structural units, each of which may have a different therapeutic target. The link between them is immolable and cleaved by various mechanisms: exposure to hydrogen peroxide, cleavage by a cysteine-dependent mechanism, hydrolysis of amide groups, etc. These entities do not have all the advantages of pleiotropic agents, since they release two active ingredients, each of which may be responsible for undesirable effects or interact with each other. Nevertheless, they make it possible to reduce the number of pharmaceutical doses and, above all, their characteristics as prodrugs confers on them an improved pharmacokinetic behaviour, such as the crossing of the blood–brain barrier, compared to that of the individual active substances, or avoid a toxicity which would otherwise be expressed. Examples include boronates releasing a derivative of tacrine, an acetylcholinesterase inhibitor, and ibuprofen, an anti-inflammatory drug (Figure 3) [10]. Ibuprofen was also associated with lipoic acid through diamino chemical linkers [9]. The strategy followed in these examples is to improve the Blood–brain barrier (BBB) permeability of the codrugs and the brain delivery of the two drugs by temporarily masking their acidic groups under ester, carbamate or amide ones. Another example concerns thio-isocyanates releasing memantine, an antagonist of *N*-methyl-D-aspartate (NMDA) receptors, inducing autophagy and neuroprotective effects, and hydrogen sulphide (H_2_S), with anti-inflammatory and anti-apoptotic properties [11]. On the other hand, benzamides were conceived to be cleaved into tacrine and probenecid, a neuroprotective agent (Figure 3) [12]. In this case, one of the objectives is to try to reduce the peripheral hepatotoxicity of tacrine which led to its withdrawing from the market. 

Some of these dissociable conjugates combine two identical structural units. Thus, they are not pleiotropic agents, unless the single drug released is itself an agent with multiple actions. This is the case, for example, for chalcones, antioxidants and inhibitors of acetylcholinesterase and lipoxygenase, which are linked by ether bridges that can be cleaved by the enzymes of cytochromes P1A1 and 1A2 [13] (Figure 3).

### 2.2. Non-Dissociable Conjugates


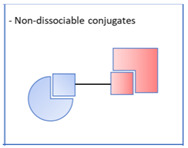
. Again, these entities are made of two structural moieties, each designed to recognise a different target [14,15,16,17,18,19,20]. However, this time, the chemical link between these elements is not cleavable. It is generally made up of aliphatic chains of variable length. These conjugates, often also called hybrids, should, in theory, be aimed at spatially close targets or at two distinct sites within the same protein. In this case, the length of the chemical bond should consider the distance separating the two therapeutic targets or the two targeted sites within a single target. This is the case, for example, with dual-site inhibitors of acetylcholinesterase (AChE), which interact with both the catalytic site of the enzyme and a peripheral site, which is described as anionic. These two sites are located on either side of an aromatic gorge within the enzyme, which is approximately 20 Å deep. For this reason, the number of atoms in the chemical bond in the structure of these dual interaction inhibitors is calculated to separate each of the two motifs by an adequate distance [21]. An example is donepezil, one of the anti-Alzheimer’s drugs currently on the market, which inhibits the catalytic site of AChE and thus temporarily alleviates the cholinergic deficiency that marks the disease, and also interacts with the peripheral anion site of the enzyme, preventing it from aggregating with the beta-amyloid peptide (Figure 4). 

Other non-dissociable conjugates do not target close locations. They are then to be considered as structural compromise that non-specifically recognise several targets (*vide infra*) with the disadvantage of generally having a high molecular weight, which is conferred by the chemical linkage they carry, and which affects, according to Lipinski’s rule of five, their bioavailability. One of these recent, non-dissociable conjugates is Memagal which combines the structural moieties of memantine and galanthamine, another AChE inhibitor, through a metabolically stable linker (Figure 5) [9]. Memagal is a nanomolar AChE inhibitor and exhibits a micromolar affinity for NMDA receptors. It displays excellent in cellulo neuroprotective effects. The same strategy, using a metabolically stable linker, was exemplified with a non-dissociable conjugate combining the structures of chloro-tacrine and tryptophan (Figure 5). The latter appears to be involved in the inhibition of β-amyloid misfolding. These conjugates, despite their unfavourable drugability profile, exhibited positive results in the reversion of scopolamine-induced cognitive deficits in rats [9]. 

### 2.3. Pleiotropic Prodrugs


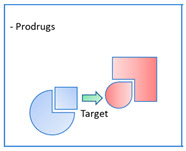
. Some multi-acting agents need to be administered in the form of prodrugs in order to overcome bioavailability problems or to mask a toxic profile. In this sense, they are not different from multi-conjugated agents, whether dissociable or not (see above), or from pleiotropic molecules making structural compromises (see below). These prodrugs are, in this case, activated by different mechanisms, such as the cleavage of sulphenamide or disulphide groups by thiol groups, such as those of glutathione [22], or the acid cleavage of Mannich bases [23] (Figure 6). These mechanisms do not involve proteins that could themselves be therapeutic targets. Pleiotropic prodrugs, on the other hand, are activated by a first therapeutic target that releases a second active molecule that is active towards a second target. This concept is relatively recent and has particular application in the treatment of neurodegenerative diseases such as Alzheimer’s disease.

Among the proteins of therapeutic interest in Alzheimer’s disease and capable of activating a pleiotropic prodrug to form an active ingredient, AChE and butyrylcholinesterase (BuChE) are prime targets. These acetylcholine-degrading enzymes can be competitively inhibited by some anti-Alzheimer drugs, such as galanthamine or donepezil. These cholinesterases can also be inhibited in a pseudo-irreversible manner, through a covalent bond between the inhibitor and the enzymes, as is the case with rivastigmine, which is currently also used against Alzheimer’s disease (Figure 6). Rivastigmine is a synthetic derivative of physostigmine, an alkaloid isolated from the Calabar bean. Chemically, it is a carbamate that exchanges this chemical group with the two types of cholinesterases, which are temporarily inhibited, with a moderate and limited therapeutic benefit in terms of the symptoms of the disease. This mechanism leads to the release of a metabolite, which does not target Alzheimer’s disease pathogenesis. This knowledge of rivastigmine’s mechanism of action led to the development of novel prodrugs that are able to release, upon inhibition of cholinesterase, an active one metabolite towards a second target of therapeutic interest. 

The most advanced pleiotropic prodrug in this context is ladostigil, a hydroxy-rasagiline carbamate (Figure 6) [24,25]. This prodrug has been shown to inhibit both AChE and BuChE, which, by decarbamylating this molecule, releases a phenol derivative of rasagiline, a monoamine oxidase B inhibitor (MAO-B), which is also used in Parkinson’s disease for its neuroprotective effect. Ladostigil has been involved in clinical trials against Alzheimer’s disease and has led to the synthesis of numerous analogues [26,27,28]. Furthermore, based on the same principle, other pleiotropic prodrugs have been designed by exploiting cholinesterase inhibition to release various phenolic drugs, such as hydroxyquinoline with metal-chelator and Aβ inhibitory properties [29] or an indole derivative with 5-HT_6_R-antagonist properties [30]. 

All these compounds have in common the structural feature of rivastigmine and more precisely its metahydroxy α-methylbenzylamine moiety. This is required to be recognised by the catalytic site of cholinesterase and confers to the inhibitor a better affinity than that of acetylcholine. The nature of the carbamate group is often similar to that of rivastigmine (*N*-methyl, *N*-ethylcarbamate) but can be modified to confer a selectivity between acetyl and butyrylcholinesterase or modulated kinetics of the enzymatic inhibition to the inhibitor [31]. Some of these modified carbamates, in addition to the initial release of a phenolic drug, are further able to liberate, upon hydrolysis of the carbamoylated enzyme, a second drug such as a melatonin [32] or amphetamine [33] derivative (Figure 6). Some of these drug candidates are currently in the pre-clinical evaluation phase for Alzheimer’s disease treatment.

### 2.4. Pleiotropic Structural Compromises


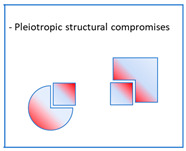
. A pleiotropic structural compromise is a molecule specifically designed to have structural elements that respond to several pharmacophores with distinct biological effects. As a result, this multi-acting entity must be able to bind indifferently to several targets and interact with each of them in such a way as to determine a synergistic therapeutic effect. The first condition for this approach to be successful is the necessary structural homology that must exist between the intended targets, so that the same drug can bind to both of them. This homology, illustrated for example by the phylogenetic tree of kinases, explains why it is possible to design pleiotropic agents that can “selectively” inhibit several kinases of therapeutic interest in oncology [34].

Numerous chimeric molecules targeting neurodegenerative diseases have also been designed based on this concept. Many of them also target cholinesterases, but generally in a non-covalent manner. Indeed, these enzymes, which, along with the NMDA glutamate receptor, are the main targets of current anti-Alzheimer’s drugs, lose their efficacy as the disease progresses and as neuronal death reduces their functionality and makes their inhibition less effective in restoring cholinergic neurotransmission. However, maintaining the integrity and functionality of cholinergic neurons longer, by means of neuroprotective or even neurotrophic agents, would theoretically make it possible to exert a symptomatic effect that would be more long-lasting over time and would be associated with an effect that would delay the progression of the disease to its severe forms. This is what is provided, for example, by donecopride, which has been specially designed to both inhibit AChE and activate the 5-HT_4_ receptor [35]. This receptor is in fact constitutively linked to an α-secretase capable of non-amyloidogenic cleavage of the neuronal membrane protein APP into various fragments, including a soluble protein known as sAPPα [36]. This protein is neurotrophic and plays an important role in neuroplasticity phenomena. This mechanism is predominant in young adults, whereas with age, this cleavage pathway is progressively abandoned in favour of an amyloidogenic pathway involving other secretases and leading, in particular, to the β-amyloid peptide. If the latter is not correctly eliminated, it will aggregate into fibrils which have been proven to be particularly neurotoxic, and then into so-called senile plaques which will constitute reservoirs and will be one of the first biomarkers of Alzheimer’s disease. It has been shown that activating the 5-HT_4_ receptor with an agonist induces the non-amyloidogenic pathway and exerts a therapeutic benefit in animal models of Alzheimer’s disease [37]. Donecopride (Figure 7) was the first pleiotropic agent with dual activity targeting both AChE and the 5-HT_4_ receptor at nanomolar concentrations, whose active sites have a certain structural homology, which allows them to be recognised indifferently by donecopride [38]. This effect, which is both pleiotropic and selective compared to other targets with no therapeutic interest, was reflected in an *in cellulo* model of Alzheimer’s disease, consisting of hippocampal neurons intoxicated by β-amyloid peptides. Donecopride targets amyloid aggregation, resulting in neuroprotection, but also hyperphosphorylation of the TAU protein, a second major biomarker of Alzheimer’s disease, and finally performs a neurotrophic action leading to synaptogenesis. Based on this profile, donecopride has also been investigated in vivo in various animal models of Alzheimer’s disease, where it has been shown to have procognitive and anti-amnesic effects and to reduce amyloid biomarkers and neuronal inflammation from an immuno-histological point of view. Donecopride and its derivatives are currently in regulatory preclinical development. 

Other activities have been associated with the competitive inhibition of cholinesterases within a unique pleiotropic molecule for therapeutic purposes in Alzheimer’s disease. As an example, propargylaminodonepezil (Figure 7) constitutes a structural compromise between donepezil and rasagiline. For this reason, it exhibits both AChE (IC_50_ = 440 nM) and MAO-B (IC_50_ = 6.4 μM) inhibitory activities [39]. Furthermore, this compound appeared to be selective towards these targets versus BuChE and MAO-A, respectively, which are less inhibited by this compound. As already mentioned for ladostigil, combining activities towards these two targets in a single compound appears clinically relevant. Propargylaminodonepezil reproduces such effects and, considering its good drugability parameters, could also make a valuable clinical candidate.

It is even possible to combine three different activities into a single structural compromise. Thereby, some indolic derivatives of donecopride (Figure 7) displayed both AChE inhibitory activity (IC_50_ = 29 nM), and 5-HT_4_R (K*_i_* = 38 nM) and σ1R affinities (K*_i_* = 5 nM) [40]. σ1R appears to be a promising target for AD treatment due to its implication in the pathogenesis of the disease and some ligands of these receptors are currently in clinical trials (vide infra), alone or in combination with drugs targeting other targets. Such indole compounds displayed interesting in vivo capacities including protect mice from dizocilpine-induced impairment in the passive avoidance test. 

In addition to AChE and the 5-HT_4_ receptor, some other triple pleiotropic drugs additionally target the 5-HT_6_ receptor, achieving the activation of one G Protein-Coupled Receptor (GPCR), while antagonising another one. Blocking 5-HT_6_R, indeed, confers antagonists or inverse agonists with procognitive effects; idalopirdine, as an example, was evaluated in phase III clinical trials [41]. Some conformationally constrained donecopride derivatives, such as phenylpyrazoles, have been specifically designed to further exert 5-HT_6_ antagonist activity (Figure 8) [42]. 

Contilisant, on the other hand, is another example of a structural compromise, since it displays both cholinesterase and MAO-A and B inhibitory effects as well histamine H_3_ receptor antagonistic properties (Figure 8) [43,44]. Contilisant is able to restore the cognitive deficit in lipopolysaccharide (LPS)-treated mice in vivo. The complex pharmacological profile of such triple pleiotropic compounds was translated into a 3D representation. Some of them displayed anti-amnesic effects in an in vivo model of scopolamine-induced deficit of working memory.

Beyond this, combining several activities within pleiotropic structural compromises has been successfully achieved in many other examples, when the intended targets have sufficient homology between them to be recognised by a single molecule. Thus, an additional antioxidant activity can be conferred to 5-HT_4_R agonists, while benzylphenylpyrrolizinones have been reported for their antioxidant, metal-chelating and amyloid β aggregation inhibitory activities (Figure 9) [45]. The latter includes some curcumin analogues with improved bioavailability parameters. 

Finally, one of the most advanced pleiotropic drug candidates against Alzheimer’s disease is Anavex 2-73 (Figure 10). This small compound acts as a dual muscarinic M1 and σ1 agonist and is able to reverse scopolamine-induced cognitive deficits in mice. Anavex 2-73 also inhibits GSK3β, oxidative stress, mitochondrial impairment and neuroinflammation [46,47,48,49]. It is currently in IIb/III phase clinical trials in Alzheimer’s patients.

## 3. Conclusions

The concept of pleiotropic compounds is currently particularly attractive to medicinal chemists who are gaining a better and better knowledge of structure–activity relationships that enables them to design “tailor-made” molecules, designed to selectively reach several targets and contributing to a synergistic therapeutic effect. This approach is often difficult, however, because it requires verification of three prerequisites that cannot be avoided. First of all, the synergistic effect of combining several drugs must be demonstrated, each of which targeting a distinct protein. To do this, it is necessary to conduct in vivo tests, proving the activity of each of the agents in relation to a given behaviour. Then, we must determine the concentrations of each of these compounds at which they lose their activity. Finally, it is necessary to check whether the co-administration of these molecules in sub-active doses allows them to recover their efficiency by synergy. Such a study has been realised by Freret et al. [50]. The authors indeed demonstrated that associating subactive doses of donepezil, a marketed drug inhibiting AChE, and RS67333, a reference 5-HT_4_R agonist, allows the recovery of the procognitive performances of these compounds that were exhibit separately at higher doses in a novel object recognition test in mice. They attested that there was a synergistic effect of the drugs, a prerequisite for the design of a unique pleiotropic drug combining these activities.

The second prerequisite consists of verifying, with chemoinformatics tools, the possibility of reaching the two selected targets by means of a single molecule, as a dissociable or non-dissociable conjugate, or a prodrug, or even a structural compromise. Such a molecular modelling study has been achieved, for example, by Sopkova et al. [51]. The authors identified common structural features to AChE inhibitors and 5-HT_4_R ligands, allowing them to build a dual pharmacophoric model, accounting for the structural requirements needed by a unique compound to exhibit both these activities. This chemoinformatic tool has been used, in addition to a structure-based drug design approach, to screen chemical libraries and to successfully select novel dual-acting compounds. 

The third prerequisite will consist, finally, in verifying the possibility of chemically constructing these entities according to the structural requirements defined by the objectives in terms of ligand–target interactions.

These preliminary steps are therefore delicate and will not always allow the possibilities of pleiotropic interactions of a single ligand with several targets. However, when this is possible, the preclinical and then clinical studies of drug candidates obtained in this way should be facilitated by the advantages of these drugs, in terms of efficacy, reduction of adverse effects, absence of drug–drug interactions and increased compliance with treatment, as previously mentioned [52]. The much higher cost of the late stages of clinical trials in the life cycle of a drug, compared with the much more modest cost of the early drug design stage, shows that it can be fruitful to spend a little more time in the design of pleiotropic drugs and explains the current interest in their development in medicinal chemistry.

## Figures and Tables

**Figure 1 pharmaceutics-15-02382-f001:**
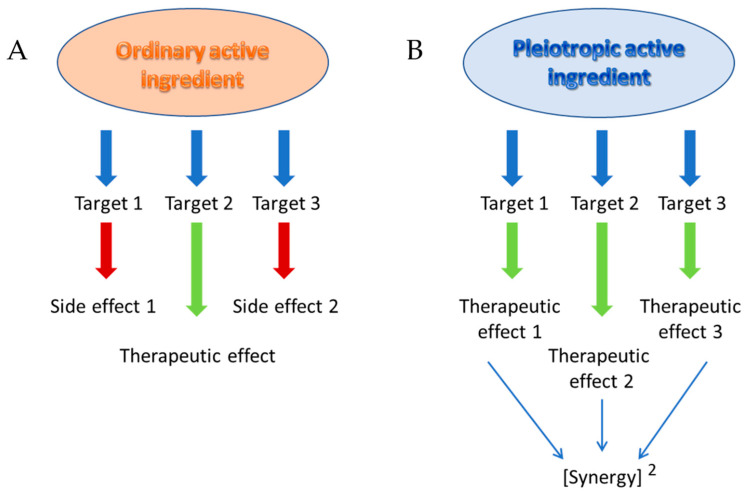
Differences between ordinary (**A**) and pleiotropic (**B**) active ingredients.

**Figure 2 pharmaceutics-15-02382-f002:**
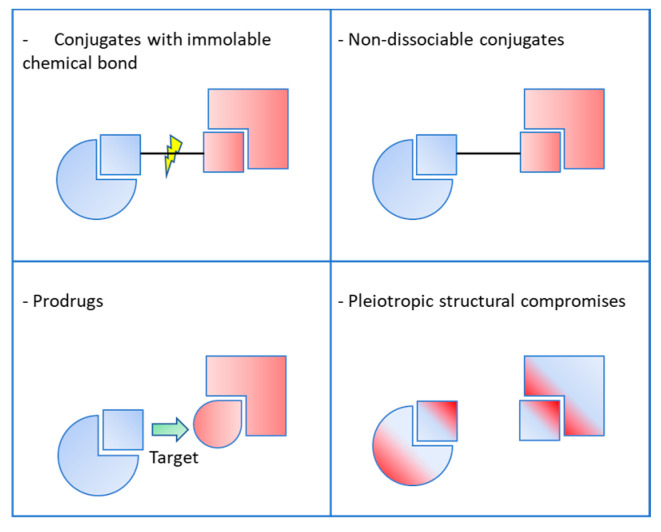
Different categories of pleiotropic compounds.

**Figure 3 pharmaceutics-15-02382-f003:**
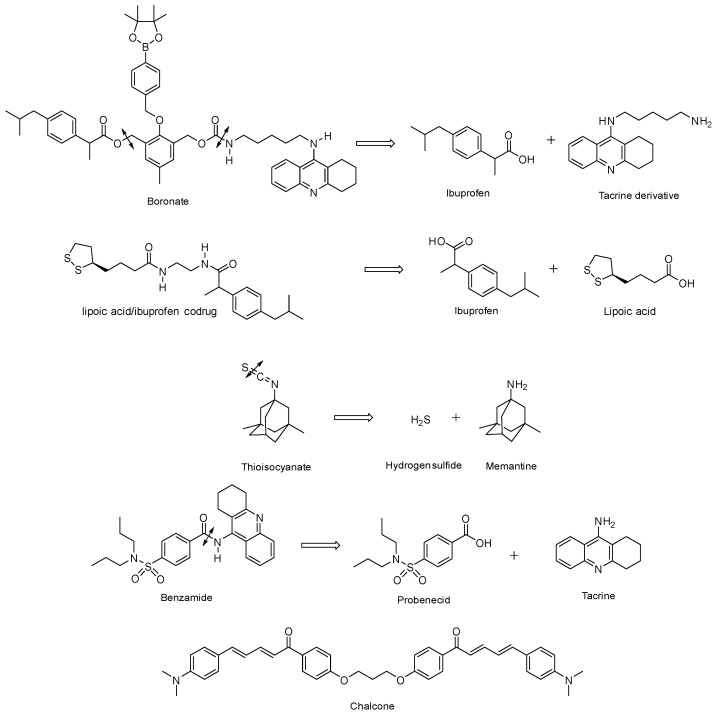
Structures of different conjugates with an immolable chemical bond.

**Figure 4 pharmaceutics-15-02382-f004:**
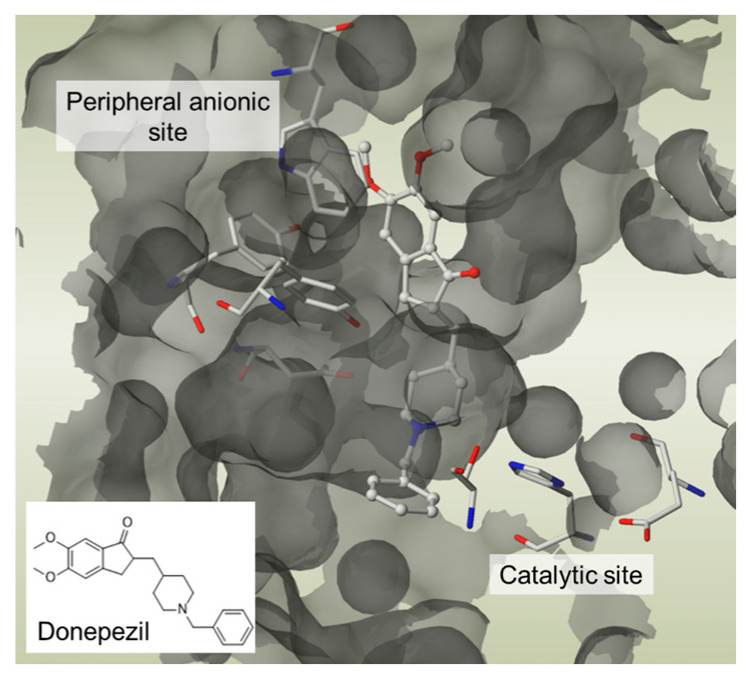
Donepezil interacts with the catalytic and peripheral anionic sites of AChE. PDB code: 1EVE. The compound and the side chains of the binding site residues are represented by the stick-like shapes. This figure was made with PYMOL (DeLano Scientific, 2002, San Carlos, CA, USA).

**Figure 5 pharmaceutics-15-02382-f005:**
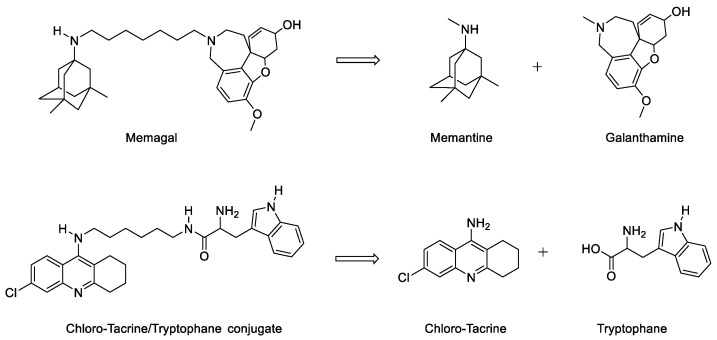
Structure of Memagal and chloro-tacrine/tryptophan non-dissociable conjugates.

**Figure 6 pharmaceutics-15-02382-f006:**
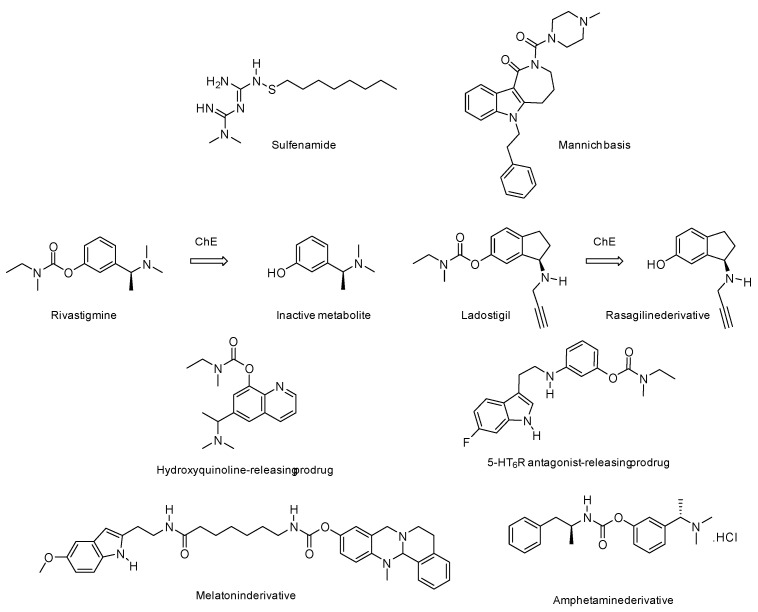
Prodrugs of pleiotropic agents.

**Figure 7 pharmaceutics-15-02382-f007:**
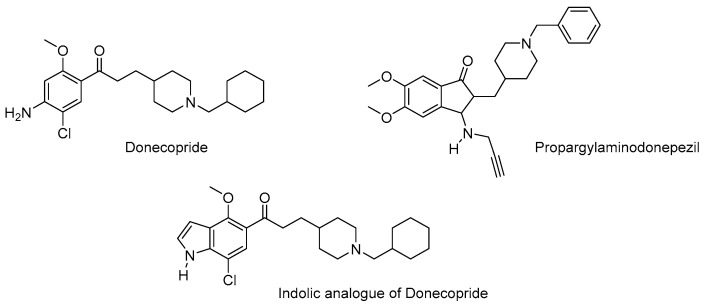
Structure of donecopride, propargylaminodonepezil and indolic analogue of donecopride.

**Figure 8 pharmaceutics-15-02382-f008:**
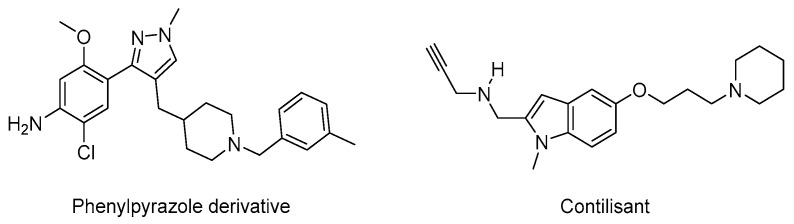
Structure of conformationally constrained donecopride analogue in phenylpyrazole series and contilisant.

**Figure 9 pharmaceutics-15-02382-f009:**
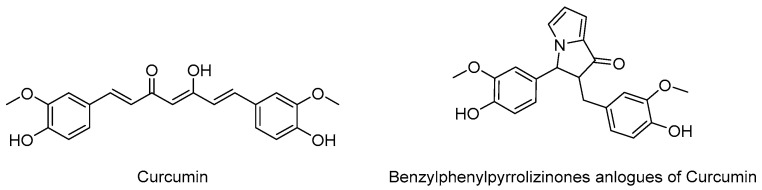
Structure of curcumin and synthetic analogues with pleiotropic activities.

**Figure 10 pharmaceutics-15-02382-f010:**
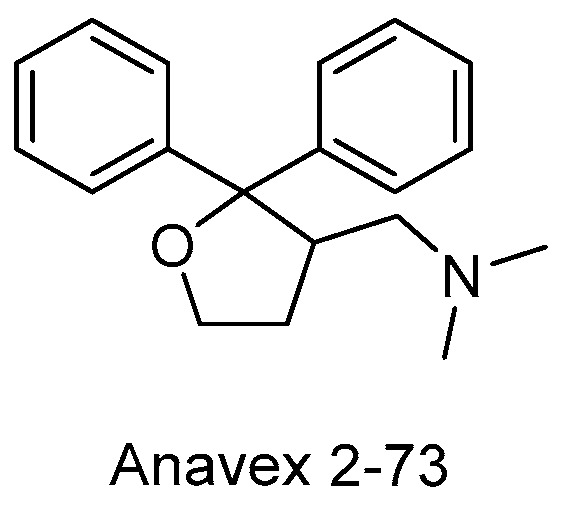
Structure of Anavex 2-73.

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
