# Peer review of "Conceptual Framework of the Design of Pleiotropic Drugs against Alzheimer’s Disease"

_pharmaceutics, 2023, doi:10.3390/pharmaceutics15102382_

Round 1

Reviewer 1 Report

This is an interesting report that needs immediate presentation to the readers. I was very pleased to serve as a reviewer. 

Author Response

The authors sincerely do want to thank the reviewer for his (her) work.

Reviewer 2 Report

The review is well organised and gives balanced and informative notion about the problem of pleiotropic drugs against Alzheimer's disease. However, the authors focused mainly on structural aspects (drug design) whether other issues (i.e., choice of targets and their combination, efficiency of certain combinations vs. single drugs, drug interaction) are just briefly mentioned or not mentioned at all. This limitation should be noted in the article with corresponding revision of the title and abstract. 

Author Response

The authors sincerely do want to thank the reviewer for his (her) work.

They agree with his (her) remark concerning the limitation of the manuscript and consequently modify the title in:

"Conceptual Framework of the Design of Pleiotropic Drugs against Alzheimer’s disease"

and the end of the abstract:

"... focusing on the structural aspect of such drug design."

The authors remain to the disposal of the reviewer for eventual other changes.

Sincerely yours

Reviewer 3 Report

This review deals with a topic of current interest on the prospect of a possible use of pleiotropic drugs in Alzheimer's disease.

the work is well written and accurately analyzes the advantages and disadvantages of using these combination of drugs from a chemical point of view.

However, the authors provide a very superficial view of Alzheimer's disease. They themselves write that it is a multifactorial disease in which it is impossible to have a single final target, but at the same time they do not argue that the problem of this disease is also the diagnosis which unfortunately occurs too late with respect to the onset and, therefore, it's to complicate to arrange a therapy also combined.

However the title does not correspond to what is then discussed in the manuscript: I would recommend for example "conceptual framework....

In the conclusions, the authors underline very well what are the still unclear points that absolutely must be investigated above all at the preclinical level.

In my opinion, this manuscript has more of the form of an editorial and not of a review which should in any case present the state of the art precisely at a preclinical level also in different murine model of AD.

Author Response

The authors sincerely do want to thank the reviewer for his (her) work.

The agree with the reviewer concerning the requisite need of early diagnosis.

Consequently they modify the sentence lines 82-86:

"...able to respond positively to treatment with pleiotropic compounds, provided that they can be administered precociously through early diagnostic remaining today to be improved. "

The authors propose also a change in the title according to the referee's recommandation:

"Conceptual Framework of the Design of Pleiotropic Drugs against Alzheimer’s disease"

The authors remain to the disposal of the reviewer for eventual other changes.

Sincerely yours

Round 2

Reviewer 3 Report

I have no further comments